# Magic123: One Image to High-Quality 3D Object Generation Using Both 2D and 3D Diffusion Priors

**Guocheng Qian**[1,2], **Jinjie Mai**[1], **Abdullah Hamdi**[3], **Jian Ren**[2], **Aliaksandr Siarohin**[2], **Bing Li**[1],
**Hsin-Ying Lee**[2], **Ivan Skorokhodov**[1], **Peter Wonka**[1], **Sergey Tulyakov**[2], **Bernard Ghanem**[1]
[1]King Abdullah University of Science and Technology (KAUST),   [2]Snap Inc.
[3]Visual Geometry Group, University of Oxford
{guocheng.qian, bernard.ghanem}@kaust.edu.sa

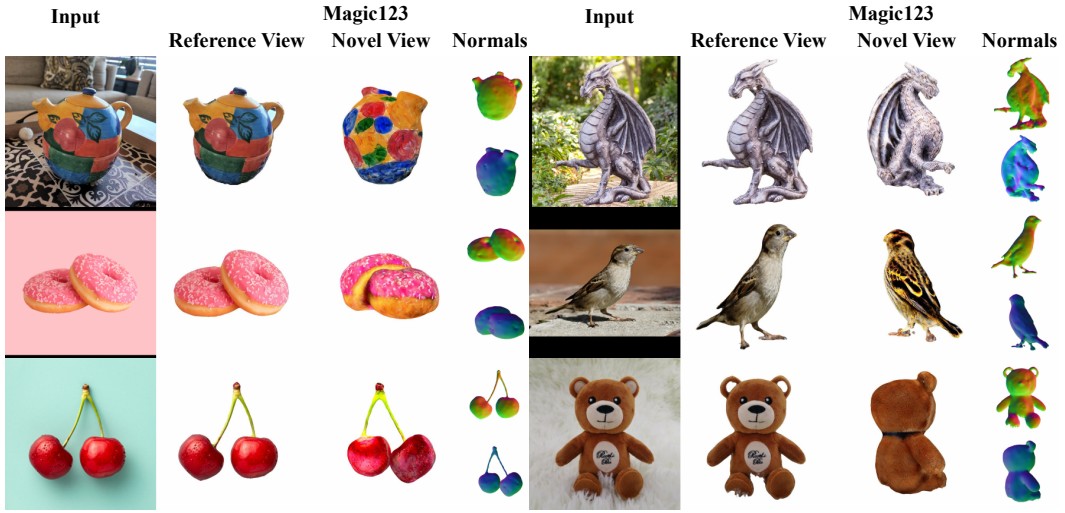

Figure 1: Magic123 can reconstruct high-fidelity 3D content with detailed geometry and high-resolution renderings (1024 × 1024) from a *single* image in the wild. Visit https://guochengqian.github.io/project/magic123/ for immersive visualizations and code.

## Abstract

We present "*Magic123*", a two-stage coarse-to-fine approach for high-quality, textured 3D mesh generation from a *single* image in the wild using *both 2D and 3D priors*. In the first stage, we optimize a neural radiance field to produce a coarse geometry. In the second stage, we adopt a memory-efficient differentiable mesh representation to yield a high-resolution mesh with a visually appealing texture. In both stages, the 3D content is learned through reference-view supervision and novel-view guidance by a joint 2D and 3D diffusion prior. We introduce a trade-off parameter between the 2D and 3D priors to control the details and 3D consistencies of the generation. Magic123 demonstrates a significant improvement over previous image-to-3D techniques, as validated through extensive experiments on diverse synthetic and real-world images.

## 1 Introduction

3D reconstruction from a single image (image-to-3D) is challenging because it is an undetermined problem. A typical image-to-3D system optimizes a 3D representation such as neural radiance field (NeRF) (Mildenhall et al., 2020), where the reference view and random novel views are differentially rendered during training. While the reference view can be optimized to match the input, there is no available supervision for the novel views. Due to this ill-posed nature, the primary focus of image-to-3D is how to leverage *priors* to guide the novel view reconstruction.

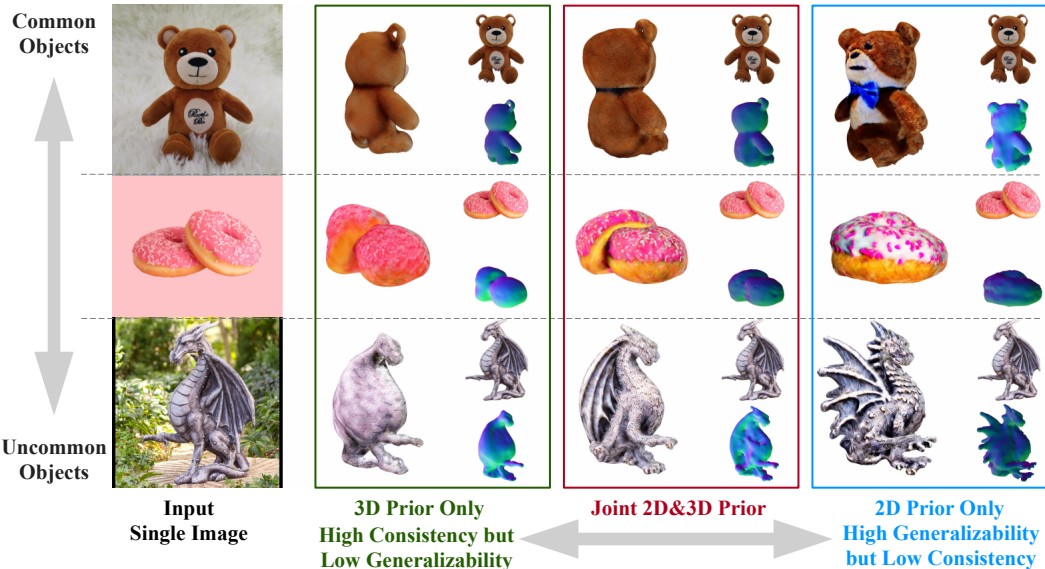

Figure 2: **The effects of the joint 2D and 3D priors.** We compare image-to-3D in three cases: a teddy bear (common object), two stacked donuts (less common object), and a dragon statue (uncommon object). Magic123 with a sole 3D prior (on the left) yields consistent yet potentially simplified 3D with reduced shape and texture details due to its low generalizability. Magic123 with a sole 2D prior (on the right) shows a strong generalizability in producing content with high details while potentially lacking 3D consistency. Magic123 proposes to use a joint 2D and 3D prior that consistently offers identity-preserving 3D with fine-grained geometry and visually appealing texture.

Current mainstream image-to-3D systems such as NeuralLift (Xu et al., 2023) and RealFusion (Melas-Kyriazi et al., 2023) employ 2D priors, *e.g.* text-to-image diffusion models (Rombach et al., 2022; Saharia et al., 2022), for 3D reconstruction. The novel views are guided by the 2D priors using text prompts associated with the input image by captioning (Li et al., 2022; 2023) or textual inversion (Gal et al., 2023). Without any 3D data, 2D prior-based solutions can distill 2D knowledge for 3D generation in a zero-shot fashion through score distillation sampling (SDS) (Poole et al., 2022). Thanks to the billion-scale training dataset (Schuhmann et al., 2021), 2D priors have been showing strong ***generalizability*** in 3D generation (Poole et al., 2022; Lin et al., 2023): successfully yielding detailed 3D content respecting various prompts. However, *methods relying on 2D priors alone inevitably compromise on 3D consistency due to their restricted 3D knowledge*. This leads to low-fidelity 3D generation, such as yielding multiple faces (Janus problems), mismatched sizes, and inconsistent texture. Fig.2 (right column) shows failure cases of using only 2D priors: the multiple faces of the teddy bear (top row) and the two donuts merged into one at the back (middle row).

Another approach to image-to-3D is to employ 3D-aware priors[1]. Earlier attempts at 3D reconstruction leveraged geometric priors like topology constraints (Wang et al., 2018) and coarse 3D shapes (Michel et al., 2022) to assist in 3D generation. However, these manually crafted 3D priors fall short of generating high-quality 3D content for various prompts. Recently, approaches like 3Dim (Watson et al., 2023) and Zero-1-to-3 (Liu et al., 2023) trained/finetuned view-dependent diffusion models and utilized them as 3D priors for image-to-3D generation. Since trained in 3D data, these 3D priors are more effective in generating content with high 3D ***consistency***. Unfortunately, (1) the scale of 3D datasets is small: the largest public dataset Objaverse-XL (Deitke et al., 2023a) only contains around 10M instances; (2) 3D datasets contain mostly limited-quality instances with simple shapes. Consequently, *3D priors are limited in generalizability and tend to generate simple geometry and texture*. As illustrated in Fig.2, while a 3D prior-based solution effectively processes common objects (for instance, the teddy bear example in the top row), it struggles with less common ones, yielding oversimplified, sometimes even flat 3D geometry (*e.g.*, dragon statue at bottom left).

---

[1]We abbreviate 3D-aware prior as 3D prior throughout the paper.

In this paper, rather than solely relying on a 2D or a 3D prior, we advocate for the simultaneous use of both priors to guide novel views. By modulating the simple yet effective tradeoff parameter between the 2D and 3D priors, we can manage a *balance between generalizability and 3D consistency* in the generated 3D content. In many cases where both 2D prior and 3D prior fail due to low 3D consistency and low generalizability, the proposed joint 2D and 3D prior propose can produce 3D content with high fidelity. Refer to Fig.2 for comparisons. **Contributions** of this work is summarized as follows:

- We introduce *Magic123*, a novel coarse-to-fine pipeline for image-to-3D generation that uses a *joint 2D and 3D prior* to guide the novel views.

- Using the exact *same* set of parameters for all examples without any additional reconfiguration, Magic123 achieves state-of-the-art image-to-3D results in both real-world and synthetic scenarios.

## 2 RELATED WORK

**Multi-view 3D reconstruction.** The development of Neural Radiance Fields (NeRF) (Mildenhall et al., 2020; Lombardi et al., 2019) has prompted a shift towards reconstructing 3D as volume radiance (Tagliasacchi & Mildenhall, 2022), enabling the synthesis of photo-realistic novel views (Barron et al., 2022). NeRF requires as many as 100 images to reconstruct a scene. Subsequent works have explored the optimization of NeRF in few-shot (*e.g.* (Jain et al., 2021; Kim et al., 2022; Du et al., 2023)) and one-shot (*e.g.* (Yu et al., 2021; Chan et al., 2022)) settings. However, these methods fail to generate 360° 3D content due to the lack of strong priors for the missing novel-view information.

**In-domain single-view 3D reconstruction.** 3D reconstruction from a single view requires strong priors on the object geometry. Direct supervision in the form of 3D shape priors is a robust way to impose such constraints for a particular domain, like human heads (Blanz & Vetter, 2003; Booth et al., 2016), hands (Pavlakos et al., 2019) or full bodies (Loper et al., 2015; Martinez et al., 2017). Such supervision requires expensive 3D annotations and manual 3D prior creation. Thus several works explore unsupervised learning of 3D geometry from object-centric datasets (*e.g.* (Kanazawa et al., 2018; Duggal & Pathak, 2022; Kemelmacher-Shlizerman, 2013; Siarohin et al., 2023)). These methods are typically structured as auto-encoders (Wu et al., 2020; Kar et al., 2015; Cheng et al., 2023) or generators (Cai et al., 2022; Sun et al., 2022) with explicit 3D decomposition under the hood. Due to the lack of large-scale 3D data, in-domain 3D reconstruction is limited to simple shapes (*e.g.* chairs and cars) and cannot generalize to more complex or uncommon objects (*e.g.* dragons and statues).

**Zero-shot single-view 3D reconstruction.** Foundational multi-modal networks (Radford et al., 2021; Rombach et al., 2022) have enabled various zero-shot 3D synthesis tasks. Earlier works employed CLIP (Radford et al., 2021) guidance for 3D generation (Jain et al., 2022; Hong et al., 2022; Mohammad Khalid et al., 2022; Xu et al., 2022) and manipulation (Michel et al., 2022; Patashnik et al., 2021) from text prompts. Modern zero-shot text-to-image generators (Ramesh et al., 2021; Rombach et al., 2022; Saharia et al., 2022; Zhang et al., 2023) improve these results by providing stronger synthesis priors (Poole et al., 2022; Wang et al., 2023a; Metzer et al., 2022; Mikaeili et al., 2023). DreamFusion (Poole et al., 2022) is a seminal work that proposed to distill an off-the-shelf diffusion model into a NeRF for a given text query. It sparked numerous follow-up approaches to improve the quality (Lin et al., 2023; Chen et al., 2023b; Wang et al., 2023b; Chen et al., 2023a). image-to-3D reconstruction (Melas-Kyriazi et al., 2023; Tang et al., 2023b; Höllein et al., 2023; Richardson et al., 2023). Inspired by the success of text-to-3D, 2D diffusion priors were also applied to image-to-3D with additional reference view reconstruction loss (Melas-Kyriazi et al., 2023; Xu et al., 2023; Seo et al., 2023; Lin et al., 2023; Raj et al., 2023). Recently, (Watson et al., 2023; Liu et al., 2023) trained pose-dependent diffusion models that are 3D-aware and used them to improve the 3D consistency. However, they suffered from low generalizability and tended to generate oversimplified geometry due to the limited quality and the small scale of 3D datasets. Our work instead explores a joint 2D and 3D prior to balance the generalizability and 3D consistency.

## 3 METHODOLOGY

We propose Magic123, a coarse-to-fine pipeline for high-quality 3D object generation from a single reference image. Magic123 is supervised by the reference view reconstruction and guided by a joint 2D and 3D prior, as shown in Fig. 3.

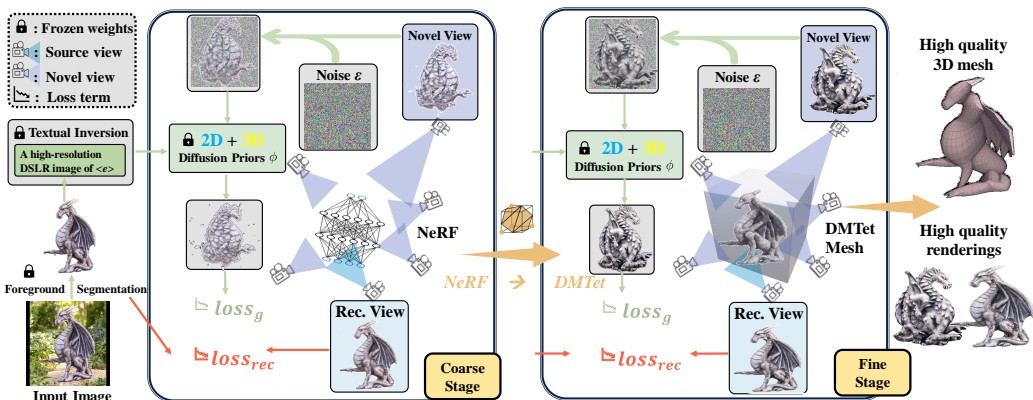

Figure 3: **Magic123 pipeline.** Magic123 is a two-stage coarse-to-fine framework for high-quality 3D generation from a single reference image. Magic123 is supervised by the reference image reconstruction and guided by a joint 2D and 3D diffusion prior. At the coarse stage, we optimize an Instant-NGP NeRF for a coarse geometry. At the fine stage, we initialize a DMTet differentiable mesh from the NeRF output and optimize it with high-resolution rendering ($1024 \times 1024$). Textural inversion is used in both stages to generate object-preserving geometry and view-consistent textures.

## 3.1 MAGIC123 PIPELINE

**Image preprocessing.** Magic123 is aimed at object-level image-to-3D generation. We leverage an off-the-shelf segmentation model, Dense Prediction Transformer (Ranftl et al., 2021), to segment the object. We denote the extracted binary segmentation mask $\mathbf{M}$. To prevent flat geometry, we further extract the depth map by a pretrained depth estimator (Ranftl et al., 2020). The foreground image is used as the input, while the mask and the depth map are used in the optimization as regularization priors.

**Coarse-to-fine pipeline.** Inspired by the text-to-3D work Magic3D (Lin et al., 2023), Magic123 adopts a coarse-to-fine pipeline for image-to-3D optimization. The coarse stage of Magic123 is targeted at learning underlying geometry that respects the reference image. Due to its strong ability to handle complex topological changes in a smooth and continuous fashion, we adopt Instant-NGP (Müller et al., 2022). It only offers low-resolution renderings ($128 \times 128$) during training because of memory-expensive volumetric rendering and possibly yields 3D shapes with noise due to its tendency to create high-frequency artifacts. Therefore, we introduce the fine stage that uses DMTet (Shen et al., 2021) to refine the coarse 3D model by the NeRF and to produce a high-resolution and disentangled geometry and texture. We use $1024 \times 1024$ rendering resolution in the fine stage, which is found to have a similar memory consumption to the coarse stage. To reconstruct 3D faithfully from a single image, we optimize the pipeline through (i) novel view guidance; (ii) reference view reconstruction supervision; (iii) two standard regularizations: depth regularization and normal smoothness.

**Novel view guidance** $\mathcal{L}_g$ is necessary to dream up the missing information. As a significant difference from previous works, we do not rely solely on a 2D prior or a 3D prior, but we leverage a joint 2D and 3D prior to optimize the novel views. See §3.2 for details.

**Reference view reconstruction loss** $\mathcal{L}_{rec}$ is to ensure the reference image $\mathbf{I}^r$ can be reconstructed from the reference viewpoint ($\mathbf{v}^r$). Mean squared error is adopted on both $\mathbf{I}^r$ and its mask as follows:

$$\mathcal{L}_{rec} = \lambda_{rgb}\|\mathbf{M} \odot (\mathbf{I}^r - G_\theta(\mathbf{v}^r))\|_2^2 + \lambda_{mask}\|\mathbf{M} - M(G_\theta(\mathbf{v}^r))\|_2^2, \tag{1}$$

where $\theta$ are the NeRF parameters to be optimized, $\odot$ is the Hadamard product, $G_\theta(\mathbf{v}^r)$ is a NeRF rendered RGB image from the reference viewpoint, $M()$ is the foreground mask acquired by integrating the volume density along the ray of each pixel. Since the foreground object is extracted as input, we do not model any background and simply use pure white for the background rendering for all experiments. $\lambda_{rgb}$ and $\lambda_{mask}$ are the weights for the foreground RGB and the mask, respectively.

**Depth regularization** $\mathcal{L}_d$ is a standard tool to avoid overly-flat or caved-in 3D content (Xu et al., 2023; Tang et al., 2023b). We would like the depth $d$ from the reference viewpoint to be similar to

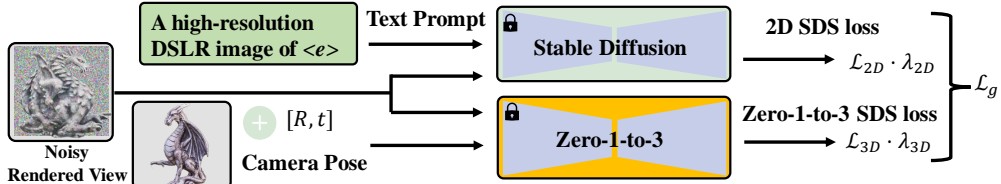

Figure 4: 2D *v.s.* 3D Diffusion priors. Magic123 uses Stable Diffusion (Rombach et al., 2022) as the 2D prior and viewpoint-conditioned diffusion model Zero-1-to-3 (Liu et al., 2023) as the 3D prior.

the depth $d^r$ estimated by a pretrained depth estimator (Ranftl et al., 2020). Due to the mismatched values of $d$ and $d^r$, we regularize them linearly through normalized negative Pearson correlation:

$$\mathcal{L}_d = \frac{1}{2}\left[1 - \frac{\text{cov}(\mathbf{M} \odot d^r, \mathbf{M} \odot d)}{\sigma(\mathbf{M} \odot d^r)\sigma(\mathbf{M} \odot d)}\right], \tag{2}$$

where $\text{cov}(\cdot)$ denotes covariance and $\sigma(\cdot)$ measures standard deviation.

**Normal smoothness** $\mathcal{L}_n$ is a common idea to reduce high-frequency artifacts. Finite differences of depth are used to estimate the normal map. Gaussian smoothness with a $9 \times 9$ kernel is applied:

$$\mathcal{L}_n = \|\mathbf{n} - \tau(g(\mathbf{n}))\|, \tag{3}$$

where $\tau(\cdot)$ denotes the stopgradient operation and $g(\cdot)$ is a Gaussian blur.

Overall, both the coarse and fine stages are optimized by a combination of losses:

$$\mathcal{L}_c = \mathcal{L}_g + \mathcal{L}_{rec} + \lambda_d \mathcal{L}_d + \lambda_n \mathcal{L}_n, \tag{4}$$

We note here that we empirically find the depth and normal regularization only have marginal affects to the final performance. We keep here as a standard practice.

## 3.2 NOVEL VIEW GUIDANCE: A JOINT 2D AND 3D PRIOR

**2D priors.** Using a single reference image is insufficient to optimize 3D. DreamFusion (Poole et al., 2022) proposes to use a 2D text-to-image diffusion model as the prior to guide the novel views via the proposed score distillation sampling (SDS) loss. SDS encodes the rendered view as latent, adds noise to it, and guesses the clean novel view conditioned on the input text prompt. Roughly speaking, SDS translates the rendered view into an image that respects both the content from the rendered view and the text. The SDS loss is illustrated in the upper part of Fig. 4 and is formulated as:

$$\nabla\mathcal{L}_{2D} \triangleq \mathbb{E}_{t,\epsilon}\left[w(t)(\epsilon_\phi(\mathbf{z}_t; \mathbf{e}, t) - \epsilon)\frac{\partial \mathbf{z}}{\partial \mathbf{I}}\frac{\partial \mathbf{I}}{\partial \theta}\right], \tag{5}$$

where $\mathbf{I}$ is a rendered view, and $\mathbf{z}_t$ is the noisy latent by adding a random Gaussian noise of a time step $t$ to the latent of $\mathbf{I}$. $\epsilon, \epsilon_\phi, \phi, \theta$ are the added noise, predicted noise, parameters of the diffusion prior, and the parameters of the 3D model. $\theta$ can be MLPs of NeRF for the coarse stage, or SDF, triangular deformations, and color field for the fine stage. DreamFusion points out that the Jacobian term of the image encoder $\frac{\partial \mathbf{z}}{\partial \mathbf{I}}$ in Eq. equation 5 can be further eliminated, making the SDS loss much more efficient in terms of both speed and memory.

**Textural inversion.** Note the prompt $\mathbf{e}$ we use for each reference image is not a pure text chosen from tedious prompt engineering. Using pure text for image-to-3D generation sometimes results in inconsistent texture due to the limited expressiveness of the human language. For example, using "A high-resolution DSLR image of a colorful teapot" will generate different colors that do not respect the reference image. We thus follow RealFusion (Melas-Kyriazi et al., 2023) to leverage the same textual inversion (Gal et al., 2023) technique to acquire a special token <*e*> to represent the object in the reference image. We use the same prompt for all examples: "A high-resolution DSLR image of <*e*>". We find that Stable Diffusion can generate an object with a more similar texture and style to the reference image with the textural inversion technique compared to the results without it.

**3D prior.** Using only the 2D prior is not sufficient to capture consistent 3D geometry due to its lack of 3D knowledge. Zero-1-to-3 (Liu et al., 2023) thus proposes a 3D(-aware) prior solution.

Zero-1-to-3 finetunes Stable Diffusion into a view-dependent version on Objaverse (Deitke et al., 2023b). Zero-1-to-3 takes a reference image and a viewpoint as input and can generate a novel view from the given viewpoint. Zero-1-to-3 thereby can be used as a strong 3D prior for 3D reconstruction. The usage of Zero-1-to-3 in an image-to-3D generation pipeline using SDS is formulated as:

$$\nabla\mathcal{L}_{3D} \triangleq \mathbb{E}_{t,\epsilon}\left[w(t)(\epsilon_\phi(\mathbf{z}_t; \mathbf{I}^r, t, R, T) - \epsilon)\frac{\partial \mathbf{I}}{\partial\theta}\right],\tag{6}$$

where $R, T$ are the camera poses passed to Zero-1-to-3. The difference between using the 3D prior and the 2D prior is illustrated in Fig. 4, where we show that the 2D prior uses text embedding as a condition while the 3D prior uses the reference view $\mathbf{I}^r$ with the novel view camera poses as conditions. The 3D prior utilizes camera poses to encourage 3D consistency and enable the usage of more 3D information compared to the 2D prior counterpart.

**A joint 2D and 3D prior.** We find that the 2D and 3D priors are complementary to each other. The 2D prior favors high imagination thanks to its strong generalizability stemming from the large-scale training dataset of diffusion models, but might lead to inconsistent geometry due to the lack of 3D knowledge. On the other hand, the 3D prior tends to generate consistent geometry but with simple shapes and less generalizability due to the small scale and the simple geometry of the 3D dataset. In the case of uncommon objects, the 3D prior might result in over-simplified geometry and texture. Instead of relying solely on a 2D or a 3D prior, we propose to use a joint 2D and 3D prior:

$$\nabla\mathcal{L}_g \triangleq \mathbb{E}_{t_1,t_2,\epsilon_1,\epsilon_2}\left[w(t)\left[\lambda_{2D}(\epsilon_{\phi_{2D}}(\mathbf{z}_{t_1}; \mathbf{e}, t_1) - \epsilon_1) + \lambda_{3D}(\epsilon_{\phi_{3D}}(\mathbf{z}_{t_2}; \mathbf{I}^r, t_2, R, T) - \epsilon_2)\right]\frac{\partial \mathbf{I}}{\partial\theta}\right]\tag{7}$$

where $\lambda_{2D}$ and $\lambda_{3D}$ determine the strength of 2D and 3D prior, respectively. Increasing $\lambda_{2D}$ leads to better generalizability, higher imagination, and more details, but less 3D consistencies. Increasing $\lambda_{3D}$ results in more 3D consistencies, but worse generalizability and fewer details. However, tuning two parameters at the same time is not user-friendly. Interestingly, through both qualitative and quantitative experiments, we find that Zero-1-to-3, the 3D prior we use, is much more tolerant to $\lambda_{3D}$ than Stable Diffusion to $\lambda_{2D}$ (see §C.1 for details). When only the 3D prior is used, *i.e.* $\lambda_{2D} = 0$, Zero-1-to-3 generates consistent results for $\lambda_{3D}$ ranging from 10 to 60. On the contrary, Stable Diffusion is rather sensitive to $\lambda_{2D}$. When setting $\lambda_{3D}$ to 0 and using the 2D prior only, the generated geometry varies a lot when $\lambda_{2D}$ is changed from 1 to 2. This observation leads us to fix $\lambda_{3D} = 40$ and to rely on tuning the $\lambda_{2D}$ to trade off the generalizability and 3D consistencies. We set $\lambda_{2D} = 1.0$ for all experiments, but this value can be tuned according to the user's preference. More details and discussions on the choice of 2D and 3D priors weights are available in Sec.4.3.

## 4 EXPERIMENTS

### 4.1 SETUPS

**NeRF4.** We use a NeRF4 dataset that we collect from 4 scenes, chair, drums, ficus, and microphone-from the synthetic NeRF dataset (Mildenhall et al., 2020). These four scenes cover complex objects (drums and ficus), a hard case (the back view of the chair), and a simple case (the microphone).

**RealFusion15.** We further use the 15 natural images released by RealFusion (Melas-Kyriazi et al., 2023) that consists of both synthetic and real images in a broad range for evalution.

**Optimization details.** We use *exactly the same* set of hyperparameters for all experiments and do not perform any per-object hyperparameter optimization. Most training details and camera settings are set to the same as RealFusion (Melas-Kyriazi et al., 2023). See §A and §B for details, respectively.

**Evaluation metrics.** Note that accurately evaluating 3D generation remains an open problem in the field. In this work, we eschew the use of a singular 3D ground truth due to the inherent ambiguity in deriving 3D structures from a single image. Instead, we adhere to the metrics employed in the most prior studies (Xu et al., 2023; Melas-Kyriazi et al., 2023), namely PSNR, LPIPS (Zhang et al., 2018), and CLIP-similarity (Radford et al., 2021). PSNR and LPIPS are gauged in the reference view to measure reconstruction quality and perceptual similarity. CLIP-similarity calculates an average CLIP distance between the 100 rendered image and the reference image to measure 3D consistency through appearance similarity across novel views and the reference view.

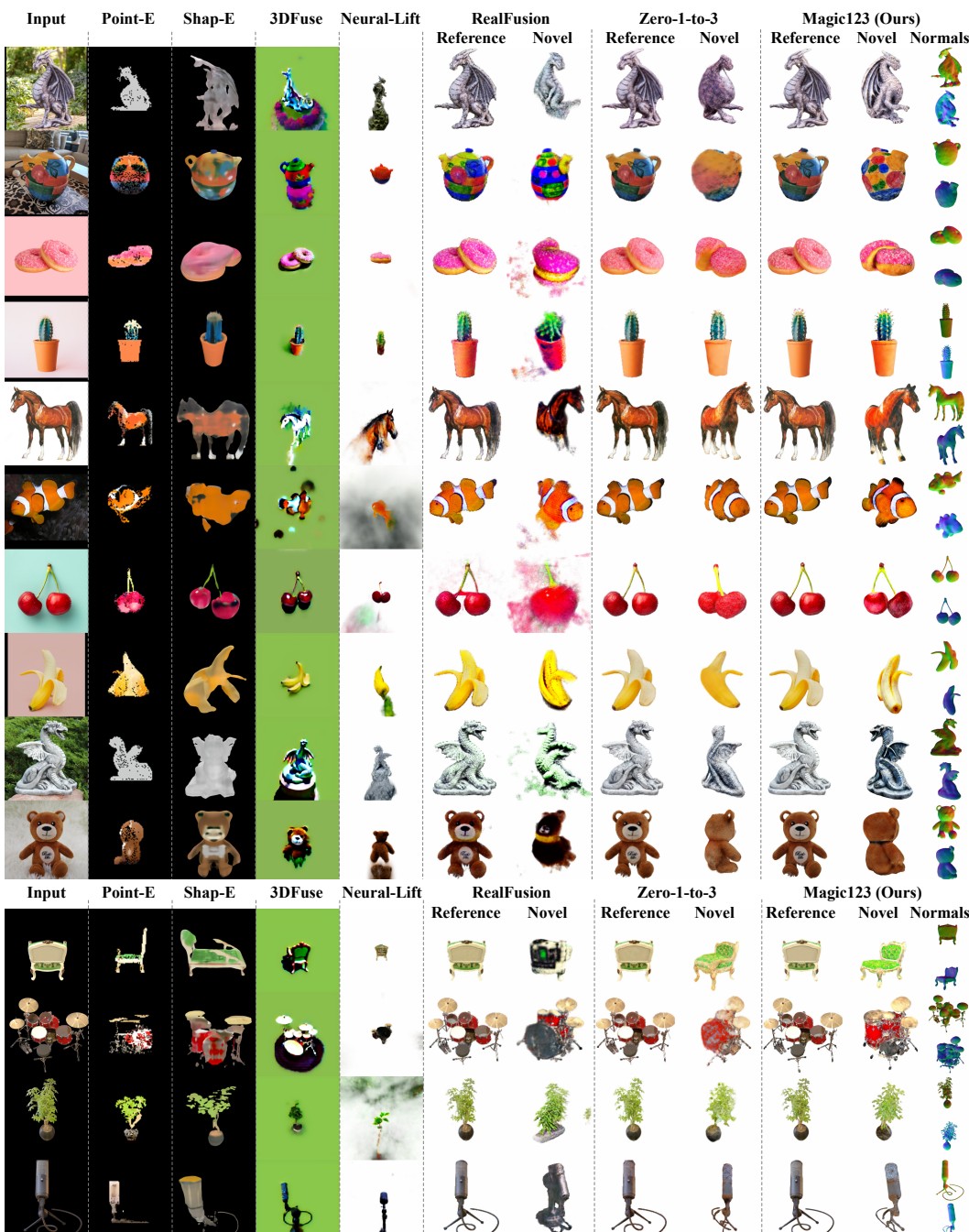

Figure 5: **Qualitative comparisons on image-to-3D generation.** We compare Magic123 to recent methods (Point-E (Nichol et al., 2022), ShapeE (Jun & Nichol, 2023), 3DFuse (Seo et al., 2023), RealFusion (Melas-Kyriazi et al., 2023), and Zero-1-to-3 (Liu et al., 2023)) for generating 3D objects from a single unposed image (the leftmost column). We show results on the RealFusion15 dataset at the top, while the NeRF4 dataset comparisons are shown at the bottom.

## 4.2  RESULTS

**Quantitative and qualitative comparisons.** We compare Magic123 against the state-of-the-art PointE (Nichol et al., 2022), Shap-E (Jun & Nichol, 2023), 3DFuse (Seo et al., 2023), NeuralLift (Xu et al., 2023), RealFusion (Melas-Kyriazi et al., 2023) and Zero-1-to-3 (Liu et al., 2023) in

Table 1: **Magic123 results.** We show quantitative results in terms of CLIP-Similarity↑ / PSNR↑ / LPIPS↓. The results are shown on the NeRF4 and Realfusion15 datasets, while **bold** reflects the best.

| Dataset | Metrics\Methods | Point-E | Shap-E | 3DFuse | NeuralLift | RealFusion | Zero-1-to-3 | **Magic123 (Ours)** |
|---|---|---|---|---|---|---|---|---|
| **NeRF4** | CLIP-Similarity↑ | 0.48 | 0.60 | 0.60 | 0.52 | 0.38 | 0.62 | **0.80** |
| | PSNR↑ | 0.70 | 0.99 | 11.64 | 12.55 | 15.37 | 23.96 | **24.62** |
| | LPIPS↓ | 0.80 | 0.76 | 0.29 | 0.40 | 0.20 | 0.05 | **0.03** |
| **RealFusion15** | CLIP-Similarity↑ | 0.53 | 0.59 | 0.67 | 0.65 | 0.67 | 0.75 | **0.82** |
| | PSNR↑ | 0.98 | 1.23 | 10.32 | 11.08 | 18.87 | 19.49 | **19.50** |
| | LPIPS↓ | 0.78 | 0.74 | 0.38 | 0.39 | 0.14 | 0.11 | **0.10** |

both NeRF4 and RealFusion15 datasets. For Zero-1-to-3, we adopt the implementation from (Tang, 2022), which yields better performance than the original implementation. For other works, we use their officially released code. All baselines and Magic123 are run with their default settings. As shown in Table 1, Magic123 achieves Top-1 performance across all the metrics in both datasets when compared to previous approaches. It is worth noting that the PSNR and LPIPS results demonstrate significant improvements over the baselines, highlighting the exceptional reconstruction performance of Magic123. The improvement of CLIP-Similarity reflects the great 3D coherence regards to the reference view. Qualitative comparisons are available in Fig. 5. Magic123 achieves the best results in terms of both geometry and texture. Note how Magic123 greatly outperforms the 3D-based zero-1-to-3 (Liu et al., 2023) especially in complex objects like the dragon statue and the colorful teapot in the first two rows, while at the same time greatly outperforming 2D-based RealFusion (Melas-Kyriazi et al., 2023) in all examples. This performance demonstrates the superiority of Magic123 over the state-of-the-art and its ability to generate high-quality 3D content.

### 4.3 ABLATION AND ANALYSIS

Magic123 introduces a coarse-to-fine pipeline for single image reconstruction and a joint 2D and 3D prior for novel view guidance. We provide analysis and ablation studies to show their effectiveness.

**The effect of the coarse-to-fine pipeline** is shown in Fig. 6. A consistent improvement in quantitative performance is observed throughout different setups when the fine stage is used. The use of a textured mesh DMTet representation enables higher quality 3D content that fits the objective and produces more compelling and higher resolution 3D visuals. Qualitative ablation for the coarse-to-fine pipeline is available in §C.2.

**Combining both 2D and 3D priors and the trade-off factor** $\lambda_{2D}$**.** Fig. 6 demonstrates the effectiveness of the joint 2D and 3D prior quantitatively. In Fig. 7, we further ablate the joint prior qualitatively and analyze the effectiveness of the trade-off hyperparameter $\lambda_{2D}$ in Eqn 7. We start from $\lambda_{2D}$=0 to use only the 3D prior and gradually increase $\lambda_{2D}$ to 0.1, 0.5, 1.0, 2, 5, and finally to use only the 2D prior with $\lambda_{2D}$=1 and $\lambda_{3D}$=0 (we also denote this case as $\lambda_{2D}$=∞ for coherence). The key observations include: (1) Relying on a sole 3D prior results in consistent geometry (*e.g.* teddy bear) but falters in generating complex and uncommon objects, often rendering oversimplified geometry with minimal details (*e.g.* dragon statue); (2) Relying on a sole 2D prior significantly

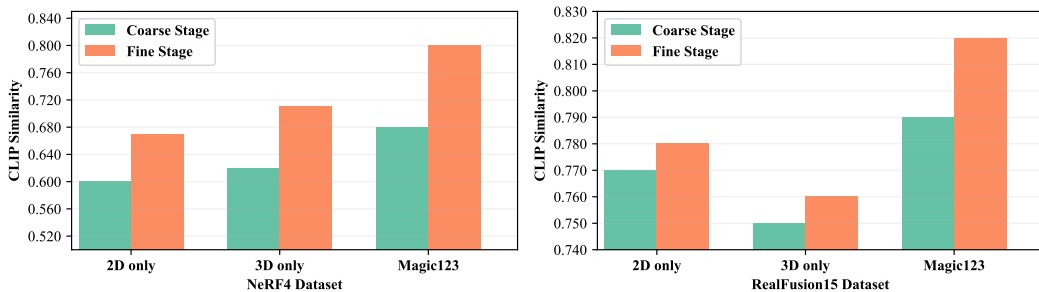

Figure 6: **Ablation study (quantitative).** We quantitatively compare using the coarse and fine stages in Magic123. In both setups, we ablate utilizing only 2D prior ($\lambda_{2D}$=1,$\lambda_{3D}$=0), utilizing only 3D prior ($\lambda_{2D}$=0,$\lambda_{3D}$=40), and utilizing a joint 2D and 3D prior ($\lambda_{2D}$=1,$\lambda_{3D}$=40).

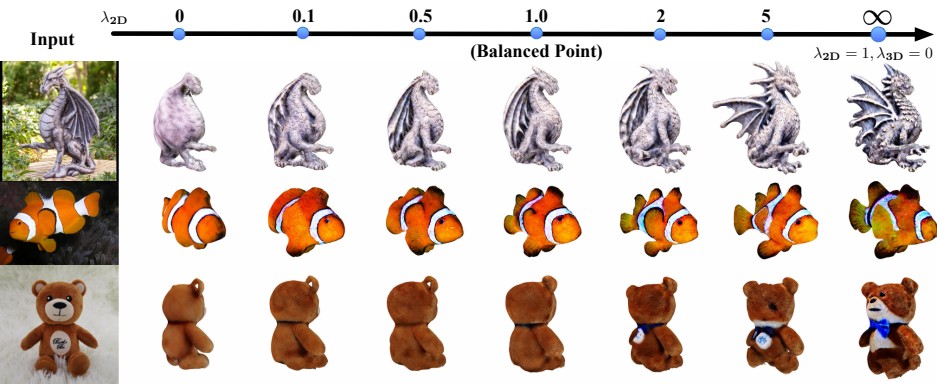

Figure 7: Setting $\lambda_{2D}$. We study the effects of $\lambda_{2D}$ on Magic123. Increasing $\lambda_{2D}$ leads to a 3D geometry with higher imagination and more details but less 3D consistencies and vice versa. $\lambda_{2D}=1$ provides a good balance and thus is used as default throughout all experiments.

improves performance in conjuring complex scenes like the dragon statue but simultaneously triggers 3D inconsistencies such as the Janus problem in the bear; (3) As $\lambda_{2D}$ escalates, the imaginative prowess of Magic123 is enhanced and more details become evident, but there is a tendency to compromise 3D consistency. We assign $\lambda_{2D}=1$ as the default value for all examples. $\lambda_{2D}$ could also be fine-tuned for even better results on certain inputs.

## 5    CONCLUSION AND DISCUSSION

This work presents Magic123, a coarse-to-fine solution for generating high-quality, textured 3D meshes from a *single* image. By leveraging a joint 2D and 3D prior, Magic123 achieves a performance that is not reachable in a sole 2D or 3D prior-based solution and sets the new state of the art in image-to-3D generation. A trade-off parameter between the 2D and 3D priors allows for control over the generalizability and the 3D consistency. Magic123 outperforms previous techniques in terms of both realism and level of detail, as demonstrated through extensive experiments on real-world images and synthetic benchmarks. Our findings contribute to narrowing the gap between human abilities in 3D reasoning and those of machines, and pave the way for future advancements in single-image 3D reconstruction. The availability of our code, models, and generated 3D assets will further facilitate research and applications in this field.

**Limitation.** One limitation is that Magic123 might suffer from inconsistent texture and inaccurate geometry due to incomplete information from a single image. Specially, a clear inconsitency appers in the boundary mostly because of blended foreground and background along the boundary segmentation errors. A texture consistency loss might be helpful. Similar to previous work, Magic123 also tends to generate over-saturated textures due to the usage of the SDS loss. The over-saturation issue becomes more severe for the second stage because of the higher resolution. See examples in Fig. 5 for these failure cases: the incomplete foot of the bear, the inconsistent texture between the front (pink color) and back views (less pink) of the donuts, the round shape of the drums, and the oversaturated color of the hoarse and the chair. Similar to other per-prompt optimization methods, Magic123 also takes around 1 hour to get a 3D model and with limited diversity. This time can be reduced through (1) replacing to Gaussian Splatting in stage 1 as shown in (Tang et al., 2023a), and (2) sampling from a small range of time steps in stage 2, following the suggestions of ICLR reviewers. The diversity issue might be possible to alleviate through VDS (Wang et al., 2023b) or training with prior guidance plus diverse 3D data.

**Acknowledgement.** The authors would like to thank Xiaoyu Xiang for the insightful discussion and Dai-Jie Wu for sharing Point-E and Shap-E results. This work was supported by the KAUST Office of Sponsored Research through the Visual Computing Center funding, as well as, the SDAIA-KAUST Center of Excellence in Data Science and Artificial Intelligence (SDAIA-KAUST AI). Part of the support is also coming from KAUST Ibn Rushd Postdoc Fellowship program.

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

## A IMPLEMENTATION DETAILS

We use *exactly the same* set of hyperparameters for all experiments and do not perform any per-object hyperparameter optimization. Both coarse and fine stages are optimized using Adam with $0.001$ learning rate and no weight decay for $5,000$ iterations. $\lambda_{rgb}, \lambda_{mask}$ are set to $5, 0.5$ for both stages. $\lambda_{2D}$ and $\lambda_{3D}$ are set to $1$ and $40$ for the first stage and are lowered to $0.001$ and $0.01$ in the second stage for refinement to alleviate oversaturated textures. We adopt the Stable Diffusion (Sohl-Dickstein et al., 2015) model of V1.5 as the 2D prior. The guidance scale of the 2D prior is set to $100$ following (Poole et al., 2022). For the 3D prior, Zero-1-to-3 (Liu et al., 2023) ($105,000$ iterations finetuned version) is leveraged. The guidance scale of Zero-1-to-3 is set to $5$ following (Liu et al., 2023). The NeRF backbone is implemented by three layers of multi-layer perceptrons with $64$ hidden dims. Regarding lighting and shading, we keep nearly the same as (Poole et al., 2022). The difference is we set the first $1,000$ iterations in the first stage to normals' shading to focus on learning geometry as inspired by (Chen et al., 2023b). For other iterations as well as the fine stage, we use diffuse shading with a probability $0.75$ and textureless shading with a probability $0.25$. The rendering resolutions are set to $128 \times 128$ and $1024 \times 1024$ for the coarse and the fine stage, respectively. For both stages, we use depth regularization and normal smoothness regularization with $\lambda_d = 0.001$ and $\lambda_n = 0.5$. Following the standard practice (Barron et al., 2021; Poole et al., 2022; Melas-Kyriazi et al., 2023), we additionally add the $0.001$ entropy regularization and $0.01$ orientation regularization in the NeRF stage. Our implementation is based on the Stable DreamFusion repo (Tang, 2022). The training of Magic123 takes roughly 1 hour on a 32G V100 GPU, while the coarse stage and the fine stage take 40 and 20 minutes, respectively.

## B CAMERA SETTINGS

**Frontal view setting.** Since the reference image is unposed, our model assumes a frontal reference view ($90°$ elevation and $0°$ azimuth) for simplicity. However, in real-world applications, these angles can be adjusted according to the input, either through intuitive estimation or camera pose detection. A simple tuning of the elevation angle can improve the benchmark performance of Magic123. For instance, in the chair example shown in Fig. II, altering the elevation angle from $90°$ to $60°$ addresses squeezed reconstruction of the chair. For research purposes, we propose to exclude this camera estimation as it does not impact the comparison between different methods and only requires engineering efforts in application.

**Rendering camera setting.** We set the camera parameters for the rendering as follows. The camera is placed $1.8$ meters from the coordinate origin, *i.e.* the radial distance is $1.8$. The field of view (FOV) of the camera is $40°$. We highlight that the 3D reconstruction performance is not sensitive to camera parameters, as long as they are reasonable, *e.g.* FOV between 20 and 60, and radial distance between 1 to 4 meters. In Fig. II, we validate that using the same camera parameters as RealFusion (Melas-Kyriazi et al., 2023) that is different from ours, *i.e.* camera radius [1.0, 1.5] and FOV [40, 70], achieves results without obvious differences as ours.

## C MORE ANALYSIS AND ABLATION STUDIES

### C.1 ABLATION AND ANALYSIS ON THE USAGE OF 2D AND 3D PRIORS

**3D priors only.** We first turn off the guidance of 2D prior by setting $\lambda_{2D} = 0$, such that we only use the 3D-aware diffusion prior Zero-1-to-3 Liu et al. (2023).

Note that Zero-1-to-3 in our paper (results in Tab. 1, Fig. 5) denotes our improved reimplemented Zeo-1-to-3 using the same training configurations as Magic123. *i.e.* Zero-1-to-3 in our paper refers to the first stage results of Maigc123 3D prior only (Fig. 2, 6, 7), where both of them use second stage DMTet fine-tuning for fair comparison.

Furthermore, we study the effects of $\lambda_{3D}$ by performing a grid search and evaluate the image-to-3D reconstruction performance, where $\lambda_{3D} = 5, 10, 20, 40, 60, 80$. Interestingly, we find that Zero-1-to-3 is very robust to the change of $\lambda_{3D}$. Tab. I demonstrates that different $\lambda_{3D}$ leads to a consistent quantitative result. We thus simply set $\lambda_{3D} = 40$ throughout the experiments since it achieves a slightly better CLIP-similarity score than other values.

Table I: **Effects** of $\lambda_{3D}$ and $\lambda_{2D}$ in Magic123 using only 2D or 3D prior on NeRF4 dataset.

| | varying $\lambda_{3D}$ when $\lambda_{2D}$=0 | | | | | varying $\lambda_{2D}$ when $\lambda_{3D}$=0 | | |
| --- | --- | --- | --- | --- | --- | --- | --- | --- |
| | *10* | *20* | *40* | *60* | *80* | *0.1* | *1* | *2* |
| CLIP-similarity↑ | 0.58 | 0.61 | 0.62 | 0.61 | 0.58 | 0.54 | 0.60 | 0.72 |
| PSNR↑ | 23.96 | 24.05 | 23.96 | 23.75 | 23.34 | 23.62 | 24.11 | 22.42 |
| LPIPS↓ | 0.04 | 0.04 | 0.05 | 0.06 | 0.08 | 0.04 | 0.04 | 0.07 |

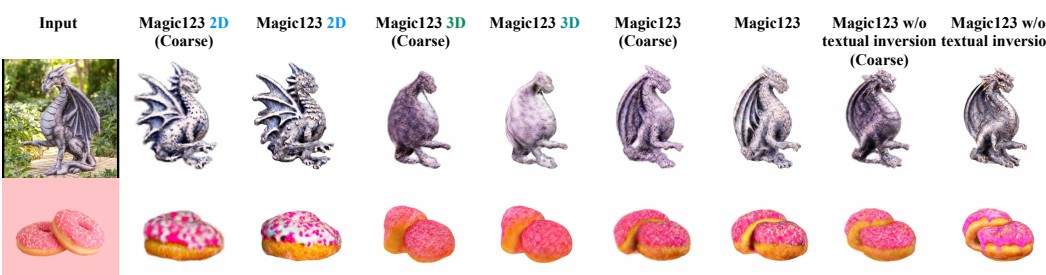

Figure I: **Ablation study (qualitative).** We qualitatively compare the novel view renderings from the coarse and fine stages in Magic123. We ablate utilizing only 2D prior ($\lambda_{2D}$=1,$\lambda_{3D}$=0), only 3D prior ($\lambda_{2D}$=0,$\lambda_{3D}$=40), and a joint 2D and 3D prior ($\lambda_{2D}$=1,$\lambda_{3D}$=40). We also ablate the effects of textual inversion at the right.

**2D priors only.** We then turn off the 3D prior and study the effect of $\lambda_{2D}$. As shown in Tab. I, the image-to-3D system is sensitive to the weights of the 2D prior. With the increase of $\lambda_{2D}$, a sharp increase in CLIP similarity and a drop in PSNR are observed. This is because a larger 2D prior weight leads to more imagination, which unfortunately might result in 3D inconsistency. Due to the observation that the 3D prior is more robust than the 2D prior to the weight, we use $\lambda_{2D}$ as the tradeoff parameter to control the imagination and 3D consistency.

## C.2  ABLATION ON THE COARSE-TO-FINE PIPELINE

In §4.3 we ablate the effect of the coarse-to-fine pipeline quantitatively. Here we provide the visual comparisons in Fig. I. The fine stage consistently augments the sharpness of the rendering and the geometry and texture details. See the edge of the wings and claws of the dragon and the toppings of the donuts for examples.

## C.3  ABLATE THE REGULARIZATION

Magic123 is optimized additionally by depth regularization, normal smoothness regularization, entropy regularization, and orientation regularization, with weights of 0.01, 0.5, 0.001, and 0.01, respectively. In Fig. II, we show that the depth, the entropy, and the orientation regularizations have minimal impact on the image-to-3D reconstruction performance. However, we keep them in our implementation as they are common practices in the NeRF family. The normal smoothness is more important in alleviating the high-frequency noise.

## C.4  ABLATE THE TEXTUAL INVERSION

We use textual inversion in both stages for consistent geometry and texture with the input reference image. Fig. I we additionally ablate the effects of textural inversion by removing it and using pure texts in the guidance. In the two examples, we change the prompts from "A high-resolution DSLR image of <e>" to "A high-resolution DSLR image of a metal dragon statue", and "A high-resolution DSLR image of two donuts", respectively. As observed, textual inversion has marginal effects on the image-to-3D reconstruction performance. However, it helps with keeping the consistency between the input image and the generated 3D content. Without textual inversion, the dragon with a different style of horns and golden textures appears that is not consistent with the input image. The donuts without textual inversion have distinct toppings from the input image.

Figure II: **Qualitative ablation study** for the effects of depth regularization, normal smoothness, entropy and orientation regularization, camera parameters, and ghe front-view assumption (elevation angle). Normal smoothness reduces high-frequency noise. Other factors like depth, entropy, and orientation regularization exert minimal influence on image-to-3D reconstruction results but are maintained in Magic123, adhering to common practice. Using different camera parameters, including camera radius (1.8 meters *v.s.* [1.0, 1.5] in RealFusion) and field of view (40 *v.s.* [40, 70] in RealFusion), have a marginal impact on performance. Magic123 opts for a simple configuration, setting the camera radius to 1.8 meters and the FOV to 40. Setting the elevation angle from 90° to a reasonable value also improves reconstruction quality (see the chair example).

## D  MORE COMPARASIONS

Here, we additionally compare Magic123 with most recent methods RealFusion (Melas-Kyriazi et al., 2023), Zero-1-to-3 (Liu et al., 2023), and Make-It-3D (Tang et al., 2023b). Magic123 outperforms all of them in terms of both 3D geometry and texture quality by a large margin.

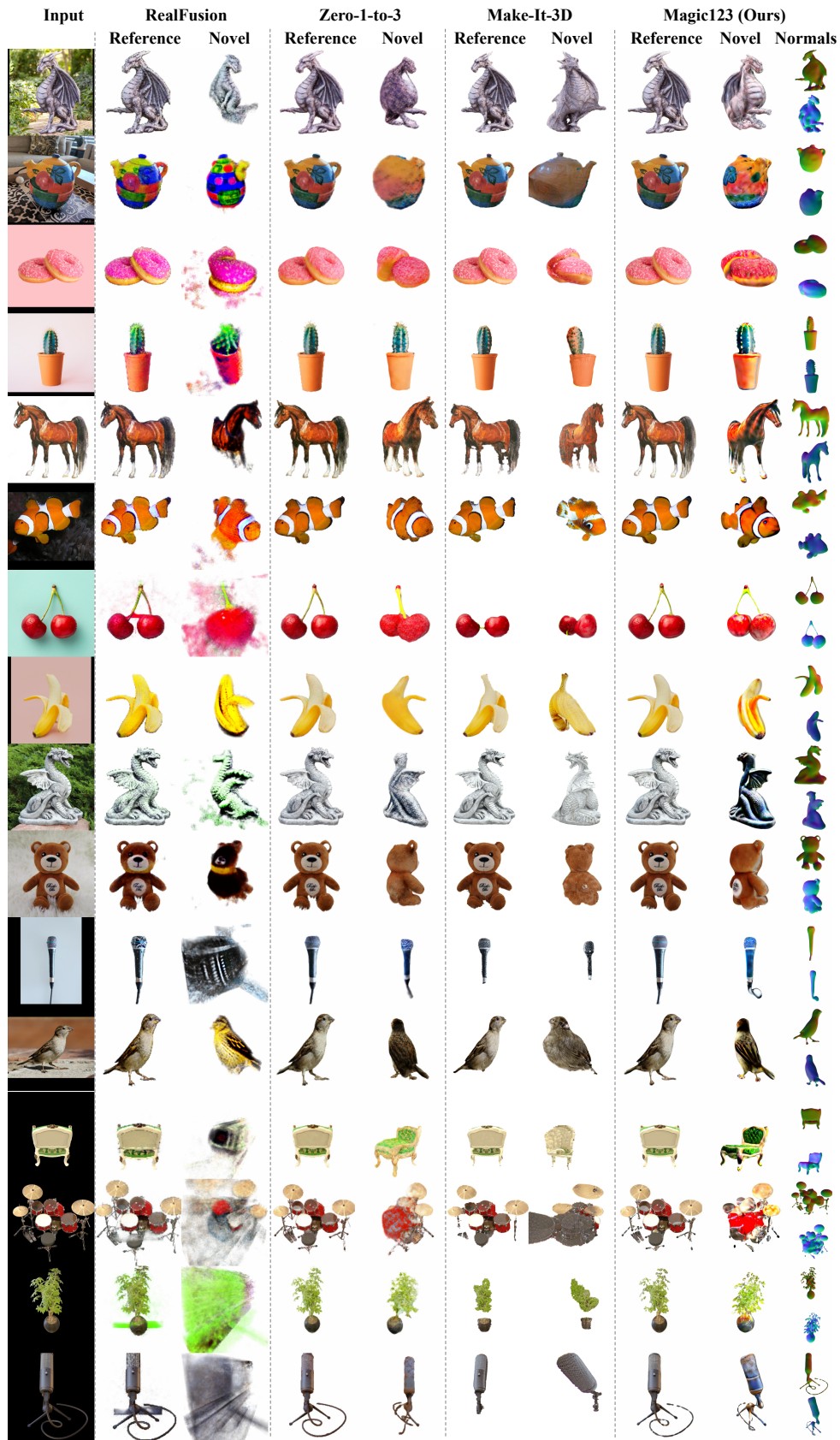

Figure III: **Qualitative comparisons.** We compare Magic123 to the most recent methods RealFusion (Melas-Kyriazi et al., 2023), Zero-1-to-3 (Liu et al., 2023), and Make-It-3D (Tang et al., 2023b).

