# OpenReview forum: "Magic123: One Image to High-Quality 3D Object Generation Using Both 2D and 3D Diffusion Priors"
_ICLR.cc/2024/Conference — ICLR 2024 poster_

### Official Review · Reviewer_jWBN · 2023-11-01

**Soundness:** 2 fair
**Presentation:** 2 fair
**Contribution:** 2 fair
**Rating:** 5
**Confidence:** 4

**Summary:**

Magic123 proposes a technique to create textured 3D meshes from a single image by a combination of both 2D and 3D priors. The method follows a two-stage strategy, transitioning from coarse to fine geometry. They also introduced trade-off parameter that attempts to balance the influences of the 2D and 3D priors, aiming to optimize quality and consistency. They showed comparisons with existing works on 3D datasets.

**Strengths:**

1. Magic123 = Magic3D and Zero123? An interesting and practical idea of introducing a balance between the 2D and 3D priors could offer a way to adjust generation outcomes.
2. The two-stage approach might offer a systematic way to enhance the granularity and accuracy of the 3D representations.

**Weaknesses:**

1. Comparison with Zero1-2-3: Although the authors claim "significant improvements over existing method", the qualitative comparison in figure 5 does not significantly support this claim. The improvements in terms of quality and sharpness are noticeable, but there seems to be a potential trade-off in consistency. Moreover, the supplementary videos are missing a direct comparison with Zero1-2-3, and the quantitative comparison with Zero1-2-3 is comparable, which further weakens the claim (in limited 3D results and examples).
2. The paper does not provide details on how the weights for the 2D and 3D priors are chosen. If these weights are determined and tuned during training and are inflexible during inference, this approach might not be the most effective way to integrate the priors.
3. The model's inherent assumption of a frontal reference limits its versatility. This is evident from the removal of four examples from the training dataset that do not meet the front view assumption. That could potentially limit the scale of 3D datasets and this could lead to even lower generalization and a drop in result quality.
4. The coarse-to-fine process, while effective, isn't novel as it has been employed in other works like Magic3D.  The Tetrahedral
representation seems to be hard to scale and appears to be optimally suited for objects. This could limit the method's applicability to a broader range of scenarios.

**Questions:**

1. The paper does not convincingly demonstrate that the combination of 2D and 3D priors leads to markedly better outcomes than using either alone. The real challenge should be to extract the unique advantages of both priors for a superior result, not just to find a balance between the two. I'd suggest the authors to add more comparison with Zero1-2-3 or include results to prove clear improvements without poor sacrifice of multi-view consistency in figure 5.
2. The paper should provide evidence of improved geometric detail and not just enhancements in the output rendering quality. Merely augmenting the contrast or enhancing the visual appeal with a 2D prior is insufficient. Could this method result in sharper geometric details as well?

---

> ### Author Response · Authors · 2023-11-20
> **Reply to Reviewer jWBN's comments on weakness**
>
> We sincerely thank the reviewer for pointing out their concerns in order to improve our paper. We address all concerns in the following. We are looking forward to a further discussion with the reviewer.
>
> Weakness 1: (1) Consistency of Zero-1-to-3 vs Magic123. We refer the reviewer to the video comparisons in supplementary files (results/human_examples.mp4, results/lambda_2d.mp4) where Magic123 outperforms Zero-1-to-3 with the same level of consistency.  (2) Compare to Zero-1-to-3. Note Zero-1-to-3 in our paper (Table 1, Figure 5) denotes our improved reimplemented Zero-1-to-3 using the same training configurations as Magic123, *i.e.* Zero-1-to-3 in our paper equals to the first stage results of Magic123 3D prior only. We also extensively compare Magic123 with Magic123 3D prior only (Figure 2, 6, 7), where both of them use second-stage DMTet finetuning for fair comparison. We will add this clarification to the paper. We would like to point out that our improved version of Zero-1-to-3 is quite a bit better than the version showcased in the original paper (See Zero-1-to-3 Fig. 9 in their original paper and compare it with Fig. 5 in our submission). Hope these address your concerns regarding the comparisons.
>
> Weakness 2:  (1) We actually provided information on how the 3D weight is chosen in the section ``A joint 2D and 3D prior'' on page 6. We mentioned when only the 3D prior is used, Zero-1-to-3 generates consistent results for $\lambda_{3D}$ ranging from 10 to 60. Thus we use a fixed value $\lambda_{3D}=40$ throughout all experiments based on trying multiple values and choosing the one that gives the best results in quantitative experiments (See Tab.I in the appendix C.1). After fixing the 3D weight, we chose the 2D weight. We find that $\lambda_{2D}=1$ gives the best quantitative results (Tab. I).  In Figure 7, we show three examples that demonstrate that choosing the 2D weight $\lambda_{2D}=1$ also gives the best qualitative results. We also tried additional examples and these other tests also confirm that $\lambda_{2D}=1$ is a good choice. For a fair automatic comparison, we therefore keep the weights fixed for all experiments at  $\lambda_{3D}=40$ and  $\lambda_{2D}=1$ without per-prompt tuning because they work generally well.
> (2) Like most other text/image-to-3D work, Magic123 is a per-prompt optimization-based method. While the above setting of $\lambda_{3D}=40$ and  $\lambda_{2D}=1$ gives good results for all examples, it is still possible to finetune the results for individual examples and possibly get slightly better results. Also, one can change $\lambda_{2D}$ to achieve a trade off between imagination and 3D consistency by tuning the weight. For example, if a user is interested in exploring more imaginative results, we recommend setting  $\lambda_{2D}$ to a higher value.
>
>
> Weakness 3: Note that the frontal view assumption is just for simplicity in showcasing the improvements due to our major contributions. We mentioned on page 15 Frontal view setting that in real-world applications, the camera angles can be adjusted according to the input, either through manual estimation or camera pose detection. We also show an example of using a more accurate camera pose and how this leads to better quality in Figure II column 5 on page 17.
>
> Weakness 4: (1) We agree with the reviewer that the usage of DMTet in the second stage is not novel by itself. However, it is a small component of our pipeline that became a standard component in 3D generation. In addition, we are the first to introduce it into the image-to-3D pipeline.  (2) We agree with the reviewer that DMTet is not optimal and it has its own advantages and disadvantages (for example scaling as pointed out by the reviewer). Despite that being said, DMTet can enhance the sharpness and significantly improve the visual quality compared to the NeRF-based representations and we therefore integrate it into our framework. Differentiable mesh representations are still an active research topic and this challenge is orthogonal to our study. In this paper, our main focus is on the usage of the joint prior, where we show its benefits through different experimental setups.

---

> ### Author Response · Authors · 2023-11-20
> **Reply to Reviewer jWBN's comments on questions**
>
> Question 1: We think this may be an overly negative interpretation of the results. We believe that our trade-off between the two priors is very attractive and that our results are generally very good. We show that the joint prior outperforms a sole prior. For example, in Figure 7 we show that Magic123 3D prior (Zero-1-to-3 + DMTet finetuning) misses details, Magic123 2D prior (SDS + DMTet finetuning) causes 3D inconsistencies, but our joint prior (Magic123) achieves better performance in terms of details and maintains consistency without sacrifice. We also would like to argue that our results are multi-view consistent at the main setting of $\lambda_{3D}=40$ and  $\lambda_{2D}=1$. Only if higher values for  $\lambda_{2D}$ are chosen there may be a sacrifice in multi-view consistency. For example, the two donuts only begin to merge into a single donut from $\lambda_{2D}=5$.
> Please refer to our video in the supplementary “results/lambda_2d.mp4” for a comparison which is a video version of Figure 7 and the example of donuts.
>
>
> Question 2: We refer to the added wings, fins, and round head in Figure 7 for reference. Another example is available in human reconstruction in the supplementary video. We compare Magic123 full with Magic123 2D prior only and Magic123 3D prior only in the scenario of human reconstruction. While Magic123 2D prior produces inconsistent 3D content and Magic123 3D prior yields simple geometry and texture, our Magic123 achieves the best quality while maintaining the 3D consistency, demonstrating the benefit of the major contribution of our work: the joint 2D and 3D prior. See the shoulder and waist geometry for the ironman example (top row), the thigh shape for the Lisa example (second row), and the back and the shoes for the Taylor Swift example (last row).

---

### Official Review · Reviewer_25qf · 2023-11-01

**Soundness:** 3 good
**Presentation:** 3 good
**Contribution:** 3 good
**Rating:** 8
**Confidence:** 5

**Summary:**

The authors propose a novel method for generating high-quality, textured 3D meshes from a single unposed image in the wild using both 2D and 3D priors. For 2D priors, the framework uses Stable Diffusion and for 3D Zero-1-to-3 is used. The approach uses two-stage coarse-to-fine pipeline that first optimizes a NeRF to produce a coarse geometry and texture, and then refines it using a memory-efficient differentiable mesh representation (DMTet) for high-resolution renderings. State of the art performance is demonstrated compared to recent image-to-3D methods.

**Strengths:**

1. **Novelty**: The proposed approach introduces a key insight of used 2D and 3D priors together to aid in better 3D geometry generation as opposed to 2D priors only as was common. Additionally, also presents the distinct advantage over 3D prior alone, since 2D diffusion models are trained on so much more data than that available to train for 3D.
2. **Paper quality**: The paper is well written with great attention to detail, all the components are described in detail and adequately motivated.
3. **Reproducibility**: The framework is implemented using available open source code. Additionally, all network and training details have been provided to aid in reproducibility of the approach. Furthermore, code has been provided in the supplementary as well to match the results shown in the paper.
4. **Result quality**: The generated assets have impressive quality for a single view lifting approach. Results are demonstrated on both synthetic objects and objects in the wild to highlight the efficiency of the approach.
5. **Related work**: An adequate treatment of the related work in the space of text-to-3D and single image-to-3D have been provided to place the given approach in the context of relevant literature.
6. **Ablations**: Key ablations have been provided, particularly, the effect of 2D only, 3D only and combined priors and demonstrating the effect of change of 2D and 3D prior weights.
6. **Supplementary materials**: The provided materials are very helpful, as it shows turntable videos of generated assets, along with several key ablations which provides a lot of insight into the different components of the approach.

**Weaknesses:**

1. **Segmentation**: The quality of the final asset is limited by the performance of the segmentation model of choice(as acknowledged by the authors). Several questions arise in this setting:
  > a. Does the diffusion priors correct for some of inaccuracies of the segmentation model?
  > b. Is it hard to reconstruct objects that are in a cluttered environment?
  > c. Is the DMTet algorithm affected by inaccurate segmentations?
  > d. What happens in instances where the segmentation algorithm outputs two or more disjoint segments, does the inductive bias of the NeRF overcome this issue? (this is potentially addressed to some extent under the limitation mentioned by the authors about handling discontinuities).
  > e. Is a segmentation network required even if the model is on a plain background? (this is potentially the case, as the mask appears in equation (1)

2. **Depth and Smoothness**: Similar to the above concern, the performance is capped by the effectiveness of the pre-trained depth and normal estimation network. This poses inherent limitations to the kind of input images that it can be applied to. Providing some insights on where these kind of estimation networks fail would be helpful. Furthermore, depth and normals potentially don't make much sense, till the NeRF has converged to a reasonably estimate of geometry.Is there a schedule associated with these losses to have an effect after initial shape has been estimated by the NeRF?

3. **Centered objects**: Owing to the nature of Zero-1-to-3, it appears that this kind of lifting works only on centered objects. What is the effect on the recovered geometry if the object is off-center but can still be segmented out accurately ?

4. **Textual inversion vs Dreambooth**: The authors mention that textual inversion is used for the 2D-SDS loss to capture some of the image specific texture information. Providing some insight on how this compares to performing Dreambooth on the model to achieve the same would be helpful.

**Questions:**

1. What is effect of applying this approach to scenes without segmentation instead of objects?
2. Is TI better than DreamBooth for 2D SDS loss?

---

> ### Author Response · Authors · 2023-11-19
> **Reply to reviewer 25qf (1/2)**
>
> We sincerely thank the reviewer for the insightful comments. We would like to address your concerns in the following:
>
> - Segmentation. Overall, a segmentation error may be present. Fortunately, the SOTA segmentation models are powerful enough in most cases. In our experiments, we also do not see any obvious drawbacks other than texture inconsistencies appearing in the segmentation boundary, as shown in our results in the paper and our supplementary videos. In the extreme case, a clean segmentation map can be obtained with humans in the loop. For your specific questions, we address them as follows:
>
> > a. Q: Does the diffusion priors correct for some of the inaccuracies of the segmentation model? A: Unfortunately, since the image-to-3D task forces the NeRF to fit the segmented reference view (eqn. 1), the usage of diffusion, which helps more in the novel views, will not fix the inaccuracy due to the reference view segmentation.
>
> > b. Q: Is it hard to reconstruct objects that are in a cluttered environment? A:  For a cluttered environment, SAM (Segment Anything Model) [1] seems to be able to do a decent job. As preprocessing is not the major part of this work and is an open problem, we do not study it in our paper. We thank the reviewer for your understanding.
>
> > c: Q: Is the DMTet algorithm affected by inaccurate segmentations? A: We witness texture inconsistencies in the segmentation boundary. This happens due to the object mixing with the background in the boundary. This error propagates in the DMTet texture and appears in the boundary of the 3D object. See our video in the supplementary file (results/magic123_results.mp4) for the boundary artifacts, \eg the black lines at the side of the microphone, the white reflections at the side of the chair, the lighter color at the side of the teddy bear. This can be solved by learning a UV map and forcing a total variance loss in the UV map. We would like to investigate more in this direction and will add this discussion to the limitation part in our final version.  We thank the reviewer for pointing out this concern.
>
> >d: Q: What happens in instances where the segmentation algorithm outputs two or more disjoint segments, does the inductive bias of the NeRF overcome this issue? A: Similar to question a, unfortunately, we force NeRF to faithfully reconstruct the output of the segmentation.  If two or more disjoint segments are generated, then NeRF will output two disjoint 3D object parts. See the cherry in Fig.5 row 7 for example.
>
> > e: Q: Is a segmentation network required even if the model is on a plain background? A: Theoretically, our algorithm should work with a plain background, where a segmentation map is not needed, or in other words, we use the identity function as the mask.  However, ideally, the current text/image-to-3D algorithms are designed for object generation instead of scene generation. Generating a plain background becomes somehow more like image-to-scene generation, which is another open problem. We would like to explore more in this interesting direction. Thank the reviewer for pointing this out.
>
>
> - Depth and normal smoothness. We thank the reviewer for pointing out this concern. We clarify that only the depth regularization requires depth estimation from the reference view. The normal smoothness is a total variance-based regularization applied to the predicted normals of the NeRF that does not require normal estimation from the reference view.  For the depth regularization, we refer the reviewer to the main paper appendix C.3 (page 16), where we find that depth regularization has minimal impact on the image-to-3D reconstruction performance. We kept it in our implementation as it is a common practice in image-to-3D. As for normal smoothness, it helps with reducing noise, as shown in appendix Figure II column 2 (page 17).
>
> - Centered objects. After an uncentered object is segmented, one can always center it in the preprocessing step. Note this centering is required due to the requirement of the camera setup in Zero1-to-3.
>
> - Textual inversion vs Dreambooth. We thank the reviewer for pointing out DreamBooth. Note that DreamBooth is mainly used for more than one sample to learn the concept from the samples. DreamBooth does not work for a single image since DreamBooth requires to finetune a diffusion model and will overfit if there is only one sample. Despite this being said, a very recent work [2] submitted to ArXiv on Oct 25th shows that it is possible to make DreamBooth work on a single image with enough data augmentation. We will investigate using DreamBooth instead of textual inversion in our final version. Thank you for this great suggestion.

---

> ### Author Response · Authors · 2023-11-19
> **Reply to reviewer 25qf (2/2)**
>
> **Questions:**
>
> Q: What is the effect of applying this approach to scenes without segmentation instead of objects?
> > A: Text-to-scene generation is out of our research focus and is an open problem. We will investigate a simple adaptation of our work to scene generation. We will keep you updated. However, we can not promise we can finish these experiments before the rebuttal deadline. We thank the reviewers for your understanding.
>
> Q: Is TI better than DreamBooth for 2D SDS loss?
> > A: As discussed above in Texture Inversion vs DreamBooth, we will compare our textual inversion to DreamBooth and data-augmented DreamBooth in our final version. We thank the reviewer for this great suggestion.
>
>
> [1] Kirillov, A., Mintun, E., Ravi, N., Mao, H., Rolland, C., Gustafson, L., Xiao, T., Whitehead, S., Berg, A.C., Lo, W.Y. and Dollár, P., 2023. Segment anything. arXiv preprint arXiv:2304.02643.
>
> [2]Sun, J., Zhang, B., Shao, R., Wang, L., Liu, W., Xie, Z. and Liu, Y., 2023. DreamCraft3D: Hierarchical 3D Generation with Bootstrapped Diffusion Prior. arXiv preprint arXiv:2310.16818.

---

> > ### Comment · Reviewer_25qf · 2023-11-22
> > **Response to queries**
> >
> > The authors do a great job of addressing most the concerns raised and provide important insights about the limitations and extent of the approach. Although the final pipeline relies on the robustness of several different networks (segmentation, depth and normal estimation) and the framework needs curated inputs for it to work reliable, I believe the insights provided in the work about  combining 2D and 3D priors is valuable and would serve as an important baseline for future works in single image lifting.

---

### Official Review · Reviewer_UQdQ · 2023-11-01

**Soundness:** 3 good
**Presentation:** 3 good
**Contribution:** 2 fair
**Rating:** 5
**Confidence:** 5

**Summary:**

This paper proposes to combine the SDS loss from both a 2D pretrained diffusion model (StableDiffusion) and a fine-tuned 3D-aware 2D diffusion model (Zero1-to-3). They show that this combination allows the generated 3D assets to have more realistic texture than using the Zero1-to-3 only, and have more plausible shapes than using the StableDiffusion only.

**Strengths:**

1. Paper is well-written and easy to follow.
2. The proposed method seems easy to reproduce, and code is attached.
3. On 4 nerf objects and 15 real objects, the proposed system outperforms baseline single-image-to-3D methods both qualitatively and quantitatively.

**Weaknesses:**

1. This seems to be a A+B style paper with majority of the components have appeared in prior works, e.g., coarse-to-fine optimization (first stage uses NeRF, second stage uses DMTet) from Magic3D, text-conditioned (using textual inversion) SDS loss from RealFusion, image-conditioned SDS loss from Zero1-to-3.

2. The novel part of this work seems the combination of the text-conditioned SDS loss and image-conditioned SDS loss. But this part might be a bit straightforward in the sense that the 3D awareness of Zero1-to-3 can help learn better geometry (as shown in Zero1-to-3), and the text-conditioned StableDiffusion can generate better appearance (as it’s a model trained on billions of real images and not tuned on the limited amount of Objaverse renderings)

3. Given single image, there're multiple plausible 3D reconstructions. This kind of diversity is missing in the evaluation part of this work.

**Questions:**

This paper is written in a clear way, and I have no further questions after reading the manuscript carefully.

---

> ### Author Response · Authors · 2023-11-19
> **Reply to Reviewer UQdQ**
>
> Dear Reviewer UQdQ,
>
> We thank you for pointing out the concern about novelty and generation diversity. We address your concerns as follows. Looking forward to your reply.
>
> - A+B style. We thank the reviewer for pointing out this concern. We acknowledge that components like two-stage training, textual inversion, and image-conditioned SDS loss, have been utilized previously. However, the novelty in our approach lies in the unique combination of 2D SDS and 3D-aware SDS losses. This integration is not a mere combination of existing methods but a synergetic synthesis that addresses specific challenges in 3D reconstruction, particularly in achieving a balance between geometry exploration and 3D consistency. Our novel joint prior achieves better performance than any single prior as validated in many experiments, and as reviewer Ea4Z stated:  “is proven to be useful in many applications.”.
>
>
> - About novelty. We acknowledge the point of view regarding novelty, but we would like to suggest that our idea may seem more obvious in hindsight than it actually is. Also, it is an important idea and we were the first to suggest it. At the beginning of the project, it was not really clear how and if the high-level idea of combining different priors could be implemented. We would also expect that a fair number of future papers will build on our idea and will combine the distillation of multiple diffusion models.  We highlight that the combination of 2D SDS and 3D-aware SDS (Zero1-to-3) losses is **not** for the balance between **geometry and texture**, **but** for the **geometry exploration** due to high imagination of 2D SDS **and 3D shape consistency** with the help of 3D-awareness from Zero1-to-3. We refer the reviewer to our video shared in the supplementary file (*results/human_examples.mp4*),  where using only 2D or only 3D results in poor geometry, our Magic123 that uses the joint prior achieves significantly better geometry.  Our novelty of using a joint 2D and 3D prior is also acknowledged by reviewer Ea4z and 25qf.
>
>
> - About diversity. We thank the reviewer for pointing out the missing diversity. Due to the mode seeking inherited from Score Distillation Sampling (SDS), all text-to-3D results look similar to each other in terms of content for the same prompt. This happens also in the image-to-3D domain. Improving the diversity of SDS is still an open problem. The concurrent work ProlicDreamer [1] proposes VSD to address this low diversity issue, but their official code has not been released yet. The reimplementation from the ThreeStudio team [2] does not support a multi-particle version of VSD yet and will have the same low-diversity issue. We believe that image-to-3D generation with high diversity is a general limitation for many papers in the research field and is not unique to our method. Many researchers identify increasing the variability as promising future work.
>
> **References**
>
> [1] Wang, Zhengyi, Cheng Lu, Yikai Wang, Fan Bao, Chongxuan Li, Hang Su, and Jun Zhu. "ProlificDreamer: High-Fidelity and Diverse Text-to-3D Generation with Variational Score Distillation." arXiv preprint arXiv:2305.16213 (2023).
>
> [2] Liu, Ying-Tian, Yuan-Chen Guo, Vikram Voleti, Ruizhi Shao, Chia-Hao Chen, Guan Luo, Zixin Zou et al. "threestudio: a modular framework for diffusion-guided 3D generation." https://github.com/threestudio-project/threestudio

---

> > ### Comment · Reviewer_UQdQ · 2023-11-19
> >
> > Thanks for the detailed response. Appreciate it! However, I'm still a bit confused: what does "the geometry exploration due to high imagination of 2D SDS" supposed to mean? Does this mean that 2D SDS loss can generate diverse 3D while 3D-aware SDS enhances 3D consistency? This seems to contradict the diversity argument you made in the response: 2D SDS lacks diversity, no?

---

> ### Author Response · Authors · 2023-11-20
> **Rely to Reviewer UQdQ about diversity**
>
> Dear reviewer UQdQ, thank you for pointing out this concern. Note we use the terminology geometry exploration rather than geometry diversity. The reason is: Magic123 achieves different geometry with increased details with the increasement of $\lambda_{2D}$. This geometry difference is not witnessed when only using 2D prior, even with different weights for the SDS loss. Also, for a certain fixed $\lambda_{2D}$ and $\lambda_{3D}$, the results of Magic123 with different random seeds will be close to each other with only minor differences.  We are running experiments and will keep you updated with the results of different random seeds. We will clarify this point in the paper.

---

> > ### Comment · Reviewer_UQdQ · 2023-11-22
> >
> > Thanks for your response. Can you clarify a bit more about "Magic123 achieves different geometry with increased details with the increasement of $\lambda_{2D}$ . This geometry difference is not witnessed when only using 2D prior"? I feel that the two sentences seem to be contradicting each other: isn't bigger  $\lambda_{2D}$ means stronger 2D prior? So stronger 2D prior means lower geometry diversity (or in your terms, geometry exploration) or higher? Thanks.

---

### Official Review · Reviewer_Ea4Z · 2023-11-02

**Soundness:** 4 excellent
**Presentation:** 4 excellent
**Contribution:** 4 excellent
**Rating:** 8
**Confidence:** 4

**Summary:**

This proposes a diffusion-based method for single-view 3D reconstruction. The key novelty is to combine 2D diffusion prior, which can achieve higher resolution more realistic appearance and 3D diffusion prior, which can achieve better 3D consistency. It designs a 2 stage optimization pipeline where the first stage uses hash grid volume representation to get the coarse reconstruction and the second stage uses DMTet representation to refine high resolution texture. Experiments on widely used datasets, such as NeRF dataset, shows significant improvements compared to prior state-of-the-arts.

**Strengths:**

1. High-quality single-view 3D reconstruction.
The proposed method shows high-quality 3D reconstruction that is significantly better than previous state-of-the-arts.

2. Comprehensive ablation studies and convincing experiments.
Authors report various benchmarks to show improvements compared to previous works, including LPIPS, PSNR, SSIM and CLIP similarities. They did ablation studies that clearly show the benefit of combining 2D and 3D diffusion priors, both qualitatively and quantitatively (Figure 6 and 7).

3. Clear novelties and technical contributions.
The idea of combining 2D and 3D diffusion prior is novel and is proven to be useful in many applications.

4. Well-written with sufficient implementation details.
This paper is well-written and easy to follow. Authors provide enough details to re-implement this paper.

**Weaknesses:**

This is a solid paper which significantly improves the baseline of single-view 3D reconstruction. The idea of using a diffusion model fine-tuned on a large-scale 3D dataset for text-to-3D or sparse-view 3D has been adopted by many recent works and the reconstruction quality has been significantly improved since then. I do not find any specific weaknesses of this paper and followings are either limitations of current works or some design choices that can improve the results.

1. Relightability.
While this work uses a diffuse shading model, the reconstructed material texture is not really relightable and will have lighting baked in. To reconstruct relightable 3D contents, we need to understand the environment lighting of the input images and use that for optimization. However, how to do that from sparse view input is still an open problem.

2. Reconstruction speed.
The reconstruction speed of the stage one may be improved by progressively increase the level of hashgrid and the resolution of the rendered image. The reconstruction speed of the second stage may be improved by sampling from a small range of time steps.

3. Backside of the object.
In some examples, the backside of the object still has a strong color shift compared to the reference views. This may be improved by using VSD loss instead of SDS loss or simply using a strong 3D diffusion prior.

**Questions:**

Overall I am very positive towards accepting this paper. I cannot see if any questions that will change my opinions so far. I will be glad to learn from authors rebuttal and other reviewers' comments.

---

> ### Author Response · Authors · 2023-11-19
> **Reply to Reviewer Ea4Z**
>
> We thank reviewer Ea4Z for highlighting our high-quality 3D reconstruction, comprehensive experiments, good novelty, and clear writing. We also sincerely thank the reviewer for pointing out the insightful open problems and the promising suggestions.
>
> - Relightability. We agree with the reviewer that single-view inverse rendering that disentangles geometry, materials, and lighting from a single input image is a very promising research topic for future work and is currently under-explored. Despite the recent progress to disentangle geometry with material in text-to-3D generation shown in Fantasia3D [1], learning lighting is still an open problem for text/image-to-3D generation.  There is some literature showing the possibility to train an illumination predictor for indoor [2] or outdoor [3] environments. Following them, we believe it is possible to finetune Stable Diffusion or ControlNet on high-quality 3D assets in different lighting conditions (can be done through Blender) to predict environment lightning from text/images conditions. Once such a model is trained, it can serve as lighting-aware prior to text/image-to-3D generation. Another possible solution is to train a big 3D reconstruction model on large-scale 3D datasets with different illumination to predict geometry, material, and lightning end-to-end.
>
> - Reconstruction speed. We thank the reviewer for this helpful suggestion. We will implement your suggestions in our codebase.
>
>
> - Backside of the object. We thank the reviewer for pointing out the color shifting. In terms of VSD loss, the official code of ProlificDreamer has not been released yet. We have tried ThreeStudio’s reimplementation, but the VSD loss is not as stable as SDS, which leads to divergent optimization for text/image-to-3D generation in some cases.  We will continue to improve Magic123 once we find a stable VSD loss to be available. In terms of a strong 3D diffusion prior, we agree this can help with color shifting. We will experiment with Zero123-XL and SyncDreamer.
>
> **References**
>
> [1] Chen, R., Chen, Y., Jiao, N. and Jia, K., 2023. Fantasia3d: Disentangling geometry and appearance for high-quality text-to-3d content creation. arXiv preprint arXiv:2303.13873.
>
> [2] Gardner, M.A., Sunkavalli, K., Yumer, E., Shen, X., Gambaretto, E., Gagné, C. and Lalonde, J.F., 2017. Learning to predict indoor illumination from a single image. arXiv preprint arXiv:1704.00090.
>
> [3] Hold-Geoffroy, Y., Athawale, A. and Lalonde, J.F., 2019. Deep sky modeling for single image outdoor lighting estimation. In Proceedings of the IEEE/CVF conference on computer vision and pattern recognition (pp. 6927-6935).

---

> > ### Comment · Reviewer_Ea4Z · 2023-11-22
> >
> > Thanks for the detailed response! My questions have been resolved!

---

### Meta-Review · Area_Chair_Y8Hj · 2023-12-07

**Metareview:**

This work proposes a hierarchical technique for 3D generation from text inputs using both base stable diffusion and zero-123 models. 2 of 4 reviewers recommend acceptance, whereas two others are slightly leaning towards rejection. Reviewers appreciated the high-quality results. Several reviewers felt that the proposed work seems like a mix of existing works and expresses concerns over the technical novelty. It is felt that the overall contributions and good result quality outweigh the weaknesses in novelty. The reviewers did raise some valuable concerns that should be addressed in the final camera-ready version of the paper, which include adding the relevant rebuttal discussions and revisions in the main paper. The authors are encouraged to make the necessary changes to the best of their ability.

**Justification For Why Not Higher Score:**

Not much technical novelty and the proposed approach is a mix of existing techniques with some useful insights.

**Justification For Why Not Lower Score:**

Two reviewers gave 8 scores and two others gave 5 scores. I do not see strong rejection points in 5-scored reviews.

---

### Decision · Program_Chairs · 2024-01-16

Accept (poster)